# C-Type Natriuretic Peptide Acts as a Microorganism-Activated Regulator of the Skin Commensals *Staphylococcus epidermidis* and *Cutibacterium acnes* in Dual-Species Biofilms

**DOI:** 10.3390/biology12030436

**Published:** 2023-03-12

**Authors:** Maria A. Ovcharova, Mikhail I. Schelkunov, Olga V. Geras’kina, Nadezhda E. Makarova, Marina V. Sukhacheva, Sergey V. Martyanov, Ekaterina D. Nevolina, Marina V. Zhurina, Alexey V. Feofanov, Ekaterina A. Botchkova, Vladimir K. Plakunov, Andrei V. Gannesen

**Affiliations:** 1Federal Research Centre “Fundamentals of Biotechnology” of Russian Academy of Sciences, Moscow 119071, Russia; 2Skolkovo Institute of Science and Technology, Moscow 121205, Russia; 3Institute for Information Transmission Problems of Russian Academy of Sciences, Moscow 127051, Russia; 4Biological Faculty, Lomonosov Moscow State University, Moscow 119192, Russia

**Keywords:** *Staphylococcus epidermidis*, *Cutibacterium acnes*, biofilms, multispecies biofilms, skin microbiota, C-type natriuretic peptide, RNA-seq, bacterial interactions in communities

## Abstract

**Simple Summary:**

Commensal (‘friendly’) microbes protect our skin from the invasion of pathogens and provide benefits for the human immune system. The microbial community exists in different microniches of the skin in the form of a biofilm—a highly organized microbial community, with properties different from traditionally studied planktonic cultures. The two most abundant skin commensals, *Staphylococcus epidermidis* and *Cutibacterium acnes,* can form biofilms inside hair follicles and sebaceous glands, and the behavior of these bacteria can maintain skin health. The study of the potential regulation of the skin microbial community (skin microbiota) by human hormones and particularly by human natriuretic peptides (NUP) has wide perspectives because of a data shortage, hormone potential to regulate the microbial community, and potential approaches in clinical practice. We used different methods to evaluate the impact of C-type natriuretic peptide (CNP) on monospecies and dual species biofilms of *C. acnes* and *S. epidermidis*, and we found that CNP boosts the competitive properties of *C. acnes* in the presence of *S. epidermidis*. Additionally, we found genes which are potential targets of CNP, hence, we made a step towards a better understanding of how peptides regulate the behavior of cutaneous bacteria populations.

**Abstract:**

The effect of C-type natriuretic peptide in a concentration closer to the normal level in human blood plasma was studied on the mono-species and dual-species biofilms of the skin commensal bacteria *Cutibacterium acnes* HL043PA2 and *Staphylococcus epidermidis* ATCC14990. Despite the marginal effect of the hormone on cutibacteria in mono-species biofilms, the presence of staphylococci in the community resulted in a global shift of the CNP effect, which appeared to increase the competitive properties of *C. acnes*, its proliferation and the metabolic activity of the community. *S. epidermidis* was mostly inhibited in the presence of CNP. Both bacteria had a significant impact on the gene expression levels revealed by RNA-seq. CNP did not affect the gene expression levels in mono-species cutibacterial biofilms; however, in the presence of staphylococci, five genes were differentially expressed in the presence of the hormone, including two ribosomal proteins and metal ABC transporter permease. In staphylococci, the Na-translocating system protein MpsB NADH-quinone oxidoreductase subunit L was downregulated in the dual-species biofilms in the presence of CNP, while in mono-species biofilms, two proteins of unknown function were downregulated. Hypothetically, at least one of the CNP mechanisms of action is via the competition for zinc, at least on cutibacteria.

## 1. Introduction

Biofilms as one of potentially three basic microbial lifestyles [1] play an important role (sometimes negative, sometimes positive) in human life. Despite lots of studies being devoted to the investigation of biofilm infections [2] and developing new antibiofilm compounds (as for instance pyrazoles [3], or graphene nanoparticles [4] or even enzymes [5]), there is still a lot which is unknown. The same is applicable to the human microbiota and its biofilms, especially in the case of its interaction with human humoral systems. In previous decades, a novel area of study has started to develop in microbiology—microbial endocrinology [6]. Human humoral factors such as hormones could now be considered regulators of microbial metabolism and behavior [6,7,8,9]. However, too little is known about the mechanisms of such interkingdom interactions. Nowadays, catecholamines are the most studied class of hormones as regulators of bacterial behavior, and some molecular mechanisms of catecholamine action have been revealed, at least for some *Enterobacteriaceae* and Firmicutes [6,10]. However, for other hormone classes, there is a shortage of information. Nevertheless, the emerging data allow for the suggestion of the same conclusions for humoral factors such as neuropeptides [7,11,12], steroids [13,14,15] and natriuretic peptides [9,16,17,18].

Natriuretic peptides (NUPs) are molecules synthetized in different organs and tissues and they are responsible for many physiological processes in the human organism such as phosphorus uptake and vascular pressure regulation, among others [19]. Their impact on the human commensal microbiota started being systematically investigated by Prof. Marc Feuilloley’s team (namely Prof. Lesouhaitier and colleagues [20]). The studies that followed showed that NUPs are able not only to affect the metabolism of a mono-species bacterial biofilm, but also to regulate the behavior of the simplest bacterial communities—dual-species biofilms [16,17,18]. Moreover, this regulatory effect may not be so easy to detect and may depend on the presence of the second counterpart in the community [18]. It was proposed that bacteria may potentially possess receptors of NUPs similar to those in humans, for instance, this was suggested for *Pseudomonas aeruginosa* [21]. The aim of the present study was to determine whether C-type natriuretic peptide (CNP) has any ability to impact the community of two the most abundant skin commensal bacteria—*Cutibacterium acnes* and *Staphylococcus epidermidis*—in concentrations lower than those studied in previous research [16]. These species inhabit the sebaceous and moist areas of the skin [22], and they are able to form biofilms in different skin microniches [23]. Due to their abundance, their role in skin homeostasis maintenance may be considered at least “important”, and the impact on their biofilms may consequently have a significant effect on the skin microbial community and skin health [23].

## 2. Materials and Methods

### 2.1. Bacterial Strains and Cultivation

Acneic strain *Cutibacterium acnes* RT5 HL043PA2 (ATCC HM-514) and standard strain *Staphylococcus epidermidis* ATCC14990 were obtained from the ATCC collection (Manassas, VA, USA). Staphylococci were stored in the tubes with 0.4% semi-liquid lysogeny broth (LB) agar (Dia-M, Moscow, Russia) covered with mineral oil (Dia-M, Moscow, Russia) at room temperature (RT). For the experiments, a sample of biomass was plated on a Petri dish with 1.5% agarized reinforced clostridial medium, as described previously [17]. Briefly, the composition of the liquid RCM medium was (g/L): yeast extract (Dia-M, Moscow, Russia), 13; peptone (Dia-M, Moscow, Russia), 10; glucose (Dia-M, Moscow, Russia), 5; sodium chloride (Dia-M, Moscow, Russia), 5; sodium acetate (Reakhim, Moscow, Russia), 3; starch (Dia-M, Moscow, Russia), 1; L-cystein hydrochloride (Biomerieux, Marcy-l′Étoile, France), 0.5; pH 7.0. Plates were incubated at 33 °C (the temperature of skin areas inhabited both by *C. acnes* and *S. epidermidis* [24,25] for 24 h, then the biomass of a colony was transferred into a 50 mL conical flask with 15 mL of liquid LB and incubated for 18–24 h to obtain the inoculum culture. *C. acnes* was stored in Petri dishes with 1.5% agar RCM medium in anaerobic zip packs (GasPack BD, Franklin Lakes, NJ, USA) with anaerobic sachets (Anaerogaz, Russia) at 33 °C. For the experiments, the biomass of the colony was transferred into a 22 mL screw-plugged glass tube filled with liquid RCM and incubated for 72 h at 33 °C. For experiments, the inocula were adjusted to an appropriate OD_540_ (0.5, 1, 2 or 4) with sterile physiological saline (PS, 0.9% NaCl).

### 2.2. Natriuretic Peptide

C-type natriuretic peptide (CNP, Alfa-Aesar, Haverhill, MA, USA), 2196.1 g/M, was dissolved in sterile MilliQ water up to the final concentration of 0.45 mM and stored at −20 °C. In the experiments, a series of stock solution dilutions in the sterile MilliQ water were prepared to obtain appropriate concentrations. 

### 2.3. Mono-Species Biofilm Growth on Polytetrafluoroethylene Cubes 

In these experiments, we tested a range of CNP concentrations from 2.7 pM (normal blood level) [26], then 27 pM, 0.27 nM and 2.7 nM. Higher concentrations were used to simulate different pathological conditions in the human organism. The method was described earlier [9]. Briefly, the biofilms of *C. acnes* and *S. epidermidis* were grown on the surface of chemically pure 4 × 4 × 4 mm polytetrafluoroethylene (PTFE) cubes in tubes filled with a liquid RCM medium. For aerobic cultivation, 3 mL of the RCM and 50 µL of the inoculum culture (OD_540_ = 0.5) were added to each tube. For anaerobic cultivation, 350 µL of an inoculum was added into 21 mL of liquid RCM in each tube. Before bacterial inoculation, an appropriate volume of CNP solution was added to each tube. Tubes without CNP served as negative controls, and tubes without bacterial inoculation were empty controls. Tubes were screw-capped and incubated for 24 h or 72 h at 33 °C and 150 rpm; then, the OD_540_ of planktonic suspensions and OD_590_ of crystal violet extracts were measured from the biofilms, as described previously.

### 2.4. Mono-Species and Dual-Species Biofilm Growth on Glass Microfiber Filters 

Glass microfiber filters (GMFF, GF/F Sigma, St. Louis, MO, USA) were used as a carrier for the biofilms, as described previously [17,18,27]. Biofilms were cultivated and obtained using the method described elsewhere [17]. Briefly, mono-species and dual-species biofilms were grown anaerobically for 24 and 72 h at 33 °C on RCM agar plates, with or without the addition of CNP. After incubation, two parameters—CFU amount and intensity of 3-(4 5-dimethyl-2-thiazolyl)-2 5-diphenyl-2h-tetrazolium bromide (MTTm Dia-M, Russia) staining—were measured to assess the growth and metabolic activity of cells in the biofilms. 

### 2.5. Study of Planktonic Cultures and Biofilm Growth Kinetics

Kinetics measurements were conducted in the same way as described previously [15,17]. Briefly, in flat-bottom 96-well microtiter plates (TPP, Trasadingen, Switzerland), mono-species and dual-species cultures and biofilms were grown in the model, with and without the initial forced pre-adhesion stage. This allowed us to distinguish mostly biofilm growth and biofilm/planktonic growth [19]. Plates were prepared anaerobically using GasPak (BD, Franklin Lakes, NJ, USA) bags, Anaerogaz (NIKI-MLT, Saint Petersburg, Russia) sachets for O_2_ removal and CO_2_ production, for an anaerobic atmosphere. To isolate the plates, a combination of modelling clay and vacuum grease was applied. Incubation was performed for 72 h at 33 °C.

Approximate kinetic parameters were calculated based on the growth curves—maximal growth rate (h^−1^), minimal generation time (h) and maximum OD_540_. As in previous studies, we assumed the initial supposition that the OD of a culture is proportional to the cell amount in suspension. Hence, the maximal growth rate was calculated by the formula µ = Ln((OD_2_/OD_1_)/(T_2_−T_1_)), where OD_1_ and OD_2_ are the OD at the beginning and end of the linear portion of the semilogarithmic plot of the growth curve, respectively. Semilogarithmic plots were built, and the linear portions were calculated using the function SLOPE in EXCEL. SLOPE allowed us to find the coefficient “a” of linear regression y = ax + b; thus, we identified the portion as “linear” when it had (i) the minimal standard deviation of coefficient “a” and (ii) “a” was maximal—the curve was closer to vertical. Deviation time t was calculated using the formula t = ln2/µ.

### 2.6. Confocal Laser Scanning Microscopy of FISH-Stained Biofilms 

The biofilms were cultivated as described in previous research [17]. Briefly, in 24-well glass-bottom plates (Eppendorf, Germany), one mL of RCM was administered per well, with or without the addition of CNP. Mono-species and dual-species biofilms were grown anaerobically for 24 h and 72 h, then the FISH was performed with the probes 5′-TCCTCCATATCTCTGCGC-3′ for *S. epidermidis* tagged with FAM, and 5′-GCCCCAAGATTACACTTCCG-3′tagged with R6G was applied for *C. acnes*. The protocol was the same as described elsewhere [17].

CLSM was performed using Zeiss LSM 510 (Carl Zeiss, Jena, Germany): a 488 nm argon laser was used for FAM-labeled samples; a 543 nm helium-neon laser was applied for R6G samples. The immersion objectives, scanning parameters, the following ZEN Blue edition image visualization and ImageJ Comstat 2 plugin “.lsm” file procession were the same as those described elsewhere [17]. The average biofilm biomass density (µm^3^/µm^2^) was calculated using the threshold 15 in the Comstat 2 plugin.

### 2.7. Study of Differential Gene Expression 

The mono-species and dual-species biofilms of *C. acnes* and *S. epidermidis* were obtained in the same protocol as was applied in the experiments with GMFFs. *S. epidermidis* biofilms were grown for 24 h, *C. acnes* dual-species biofilms were grown for 72 h to obtain sufficient *C. acnes* for correct RNA extraction. The total RNA extraction, quality control and storage were performed as described previously [27]. Briefly, after the glass-shredding destruction of cells frozen with liquid nitrogen and Qiagen RNeasy^®^ Mini Kit (Qiagen, Hilden, Germany) application for RNA isolation, electrophoresis was performed in 1% agarose gel with the addition of ethidium bromide embedded in a standard TAE buffer to prove the correct RNA extraction (by visualization of two stripes of ribosomal RNA). Then, the isolates were stored at −80 °C before sequencing.

The concentration of RNA in the samples was measured with Qubit 2.0 (Invitrogen, Waltham, MA, USA). Ribosomal RNA depletion was conducted using the Illumina Ribo-Zero Plus rRNA Depletion Kit (Illumina, San Diego, CA, USA). Depletion was performed according to the manufacturer’s protocol with 120 ng of total RNA for each sample. Next, RNA libraries were prepared using the NEBNext Ultra™ II Directional RNA Library Prep Kit for Illumina (New England Biolabs^®^ Inc., Ipswich, MA, USA) according to the manufacturer’s protocol. RNA was fragmented for 5 min. Libraries were indexed using the index primers set NEBNext Multiplex Oligos for Illumina (Dual Index Primers Set 2) from New England Biolabs^®^ Inc., USA. Library amplification was carried out in 15 PCR cycles. Reads were generated by bcl2fastq 2.20 (https://support.illumina.com/sequencing/sequencing_software/bcl2fastq-conversion-software.htmL (accessed on 22 June 2017)) without allowing mismatches in sequencing indexes (“--barcode-mismatches = 0″).

### 2.8. Read Preprocessing and Quality Control

Actual versions of programs used in the read processing are summarized in Table 1.

Read trimming was performed in Trimmomatic [28]. Reads were aligned to reference genomes (for *C. acnes* the assembly accession № in the RefSeq database was GCF_000144755.1; for *S. epidermidis* it was—GCF_002087975.1) using BWA [29] with the BWA-MEM algorithm. The number of reads belonging to different genes were calculated using Salmon 1.3.0 [30] with 20 Gibbs samples and a correction for GC bias. Different samples were cultivated in different years to account for a possible batch effect related to the year of cultivation. A batch effect correction was performed using the ComBat-seq algorithm as a part of sva library in R. Then, the differential expression was analyzed using DeSeq2 1.22.2 with the default parameters. Hierarchical clustering was performed using the hclust function of the R programming language with the complete linkage clustering algorithm. A principal component analysis was performed using the plotPCA function of DeSeq2.

To analyze the level of contamination in reads, 1000 reads from each library were aligned by BLASTN 2.11.0 [31] to the NCBI nt database with a maximum e-value of 10^−5^. The taxonomy of the best BLAST hit according to the NCBI Taxonomy database was used to infer the taxonomy of the read source. The NCBI nt and NCBI Taxonomy databases were current as of 20 April 2022. The analysis showed that the level of contamination was negligible (Appendix A).

During the alignment of reads of dual-species biofilms, a joint reference was used which was formed by combining the reference genomes of *C. acnes* and *S. epidermidis* (accession codes provided above). After the read alignment, information on reads that aligned to different genomes was split and used for differential analysis independently.

### 2.9. Quantitative PCR

To approve the RNA-seq analysis results for differential gene expression, qPCR was performed for mono-species and dual-species biofilms of *C. acnes* and *S. epidermidis*. The total RNA extraction was conducted as described previously in at least five independent repeats. For the dual-species communities, part of the samples was obtained in May 2022, and the last part was obtained in August–September 2022 (10 independent repeats in total). Primers for each gene revealed as differentially expressed were synthetized by Evrogen (Russia) and tested for their acceptability, including the absence of cross-species hybridization and autohybridization products. For the primer test, the total DNA was extracted from cultures of each bacterium. The cells were disrupted, being frozen with liquid nitrogen as described above, then Promega Wizard^®^ Genomic DNA Purification Kit (Promega, Madison, WI, USA) was applied to isolate the DNA. The DNA was stored at −20 °C and the PCR was conducted for a primer test using the Syntol (Moscow, Russia) PCR kit (Table 2). The primers used for qPCR are listed in the Appendix A.

For qPCR, the synthesis of the first chain of cDNA was conducted using the MMLV Reverse Transcriptase Kit (Evrogen, Russia) in accordance with the manufacturer’s protocol. Then, qPCR was performed in the PCR buffer-PB (Syntol, Moscow, Russia) in the presence of SYBR Green I and passive reference dye ROX for fluorescence signal normalization. Detection for each sample was conducted in duplicate. The ddH_2_O (Syntol, Moscow, Russia) was used as a negative control. Amplification was performed using the PCR System CFX96 Touch TM (Bio-Rad, Hercules, CA, USA). The temporal and temperature profile of the reaction was: polymerase activation 5 min at 95 °C; then 40 cycles of 15 s at 95 °C, 20 s at 55 °C and 40 s at 62 °C. The gene differential expression was measured in relation to the control samples (biofilms grown without CNP) as a calibrator. As a reference, genes of 16S rRNA of S. epidermidis and C. acnes were selected where applicable (for dual-species communities, both genes were used). The amount of a target gene normalized to the internal control and to a calibrator was identified using the comparison method Ct (ΔΔCt) as 2^−ΔΔCt^, as described previously [27].

### 2.10. In Silico Protein Sequence Analysis

Protein homolog searches, alignment analyses and building similarity trees were performed using NCBI BLAST tools (https://blast.ncbi.nlm.nih.gov/Blast.cgi (accessed on 10 July 2022)), the NCBI protein database (https://www.ncbi.nlm.nih.gov/protein (accessed on 15 July 2022)) and the UniProt database (https://www.uniprot.org/ (accessed on 15 July 2022)). 

### 2.11. Microbiological Statistics

All of the experiments were conducted at least in triplicate. In the microbiological experiments, where applicable, the data were displayed as histograms with medians as bar charts, where all the data values were placed as dots and the error bars spanning the maximal and the minimal points. All the microbiological data were processed in GraphPad Prism or in Microsoft EXCEL 2019. All the plots (except the RNA-seq) were built in GraphPad Prism.

Statistical analysis of the data (except the RNA sequencing described above) was performed using the nonparametric Mann–Whitney U test. In the RNA-seq experiments, *q*-values were calculated from the *p*-value in each experiment using a false discovery rate correction, as proposed previously [32]. The q-values are indicated on the data plots; q-values and *p*-values are displayed on the plots where applicable. 

## 3. Results

### 3.1. Biofilm Growth on PTFE Cubes

We tested CNP in different concentrations and found that the hormone had an impact on the biofilms of both bacteria (Figure 1 and Figure 2). As was shown previously [17,18], the hormone mostly affected biofilm growth and had no impact on planktonic cultures (Figure 1A,B and Figure 2A,B). After 24 h of incubation of *S. epidermidis* biofilms, we observed an inhibitory effect at all CNP concentrations except 27 pM. Interestingly, this was most pronounced at the “physiological” concentration of 2.7 pM, 78.8% (Figure 1C). In the case of *C. acnes*, after 24 h of incubation, there was a slight tendency towards stimulation. After 72 h, *S. epidermidis* biofilms were stimulated in the presence of CNP at lower concentrations (2.7 pM, 118.1%; 27 pM,127.7%; Figure 1D). 

*C. acnes* was slightly inhibited at those concentrations, but the variability of data was too high (Figure 2C,D). Data variability is a common phenomenon for CV staining [33,34], as referred to in our previous works with *C. acnes*. Nevertheless, based on the reaction of *S. epidermidis* on CNP, we used a 2.7 pM concentration in the following experiments.

### 3.2. Biofilms on Glass Microfiber Filters

First, we analyzed how CNP affects the intensity of MTT reduction in mono-species and dual-species biofilms (Figure 3). We found that, in mono-species *C. acnes* biofilms, in control samples after 72 h, the metabolic activity was more than two-times higher than after 24 h at 3.38 and 1.18, respectively. The addition of 2.7 pM CNP resulted in a slightly higher median formazan OD after 24 h of incubation and this was lower after 72 h of incubation. No significant changes were found in *C. acnes*. In the case of *S. epidermidis*, in controls, the median OD_540_ was 4.19 after 24 h and 2.1 after 72 h of incubation. This is logical due to staphylococci being fast growing bacteria and after 72 h they form a mature biofilm, where cells reduce their activity. The addition of CNP reduced the OD in the 24 h samples and increased the OD in the 72 h samples, but this was not significant. Hence, the hormone seemed to have the opposite effect on the two bacteria, but this was not statistically significant. 

In dual-species biofilms, we found that the OD_540_ was virtually the same at both 24 h and 72 h (4.3 and 3.8, respectively); however, both after 24 h and 72 h, the OD was less than the sum of OD_540_ of the corresponding mono-species biofilms. Hence, the competitive interactions between bacteria resulted in the final metabolic activity in the community. Moreover, those interactions significantly changed the mode of action of CNP. After 24 h of incubation, there was strong decrease in OD_540_ to 2.55 in the presence of CNP. After 72 h of incubation, there was slight but statistically significant decrease in OD_540_ to 3.4. Hence, on the one hand, after 24 h of incubation, *C. acnes* increased the inhibitory effect of CNP on *S. epidermidis*, the dominant bacterium in the community [17]. On the other hand, after 72 h of incubation, the community behavior was closer to the 72 h *C. acnes* mono-species biofilms, and the hormone decreased the metabolic activity rather than stimulating it, as in *S. epidermidis* mono-species biofilms.

The analysis of CFU quantities showed the differences between mono-species and dual-species biofilms (Figure 4). After 24 h of incubation, in mono-species *S. epidermidis* biofilms, no effect on CNP was observed. However, despite the large variability of data, the median CFU counts of *S. epidermidis* biofilms were less than half, compared with the control (6.8 × 10^9^ and 2.8 × 10^9^, respectively, Figure 4A). After 72 h of incubation, the CFU count in the *S. epidermidis* mono-species biofilms was similar to those in the 24 h samples (from 3 × 10^9^ to 6 × 10^9^), and no effect of the hormone was observed. However, the *S. epidermidis* CFU quantity in dual-species biofilms decreased significantly to 5 × 10^8^ in the control and 2.5 × 10^8^ in the presence of CNP (Figure 4B). This could be explained by changes in aggregation, as we observed in the case of A-type natriuretic peptide (ANP) previously [17]. Hence, the tendency towards a CFU decrease in dual-species biofilms in the presence of CNP was observed both in the 24 h and 72 h biofilms, despite the similarly large variability of data.

In the case of 24 h *C. acnes* biofilms, there was decrease in CFU in dual-species biofilms from 2.5–4.5 × 10^8^ to 2.5–3.5 × 10^6^ (Figure 4C). Despite the high variability and no statistical significance for differences between the controls and CNP samples, the tendency was controversial. In the case of *S. epidermidis*, the hormones increased the *C. acnes* CFU both in mono-species and dual-species biofilms. After 72 h of incubation, the *C. acnes* CFU in mono-species biofilms was 1.2 × 10^10^ in the controls and 5 ×10^9^ in the presence of CNP. In dual-species controls, it was reduced significantly to 4.5 × 10^7^; however, in the presence of CNP, it was increased back to 1 × 10^9^ (Figure 4D). Hence, despite the absence of a calculated statistical significance of such a difference, we could propose a strong increase in *C. acnes* CFU in dual-species mature biofilms in the presence of CNP.

### 3.3. Confocal Microscopy of Biofilms 

The analysis of images shows that, in glass-bottom plates, *S. epidermidis* was not sensitive to CNP (Figure 5A). However, there was a significant increase in biomass density in *C. acnes* mono-species biofilms after 24 h of incubation from 0.001 μm^3^/ μm^2^ to 0.002 μm^3^/ μm^2^ in the presence of CNP (Figure 5B). After 72 h of incubation, there was also a stimulatory effect of CNP (from 0.004 to 0.043 μm^3^/ μm^2^); however, the Mann–Whitney U test did not allow for the consideration of the significance of the difference due to data variability. Nevertheless, the stimulative tendency was shown for all the samples.

In dual-species biofilms, after 24 h of incubation, the density of the *S. epidermidis* biomass did not change significantly, either in presence of CNP or in comparison with mono-species biofilms; the medians were between 3.18 and 3.3 μm^3^/ μm^2^ in all four groups of samples. Nevertheless, after 72 h of incubation of the control samples, there was a decrease in biomass density to 0.93 μm^3^/ μm^2^ in comparison with the 24 h samples. The addition of CNP led to a decrease in the median biofilm density to 0.07 μm^3^/ μm^2^; however, the data variability allowed us to only establish the tendency instead of statistically significant differences (Figure 5C). In the case of *C. acnes* after 24 h of incubation, there was an increase in biomass density from 0.001 to 0.081 μm^3^/ μm^2^. After 72 h of incubation, there was a decrease in *C. acnes* biomass density from 0.007 to 0.002 μm^3^/ μm^2^ (Figure 5D). This seems to be related to a decrease in biomass of *S. epidermidis*, which is a dominant part of the community and provides the general structure of a dual-species biofilm. Hence, the decrease in the staphylococcal proportion led to a decrease in *C. acnes* biomass.

The last experiment was undertaken by the analysis of the bacterial biomass ratio in dual-species communities (Figure 6). Here, the ratio of *S. epidermidis* and *C. acnes* biomass shifted, after both 24 h and 72 h of incubation, to *C. acnes* when CNP was added. After 24 h of the control, the median *C. acnes* proportion was only 0.03% of the whole community biomass. In the presence of CNP, it became 3.22%. In the 72 h control samples, the *C. acnes* proportion was 0.81%; however, in the presence of CNP, it became 2.98%. Hence, cutibacteria were stimulated by CNP in the community in a similar way as was recently shown for ANP [17]. The confocal images are presented in the Appendix A.

### 3.4. Study of Growth Kinetics

Analysis of bacterial mono-species and dual-species cultures in systems with and without forced adhesion revealed some interesting facts. First, the adhesion stage was an important factor for growth rate; in systems without initial forced adhesion, planktonic cultures dominated and the growth rate in mono-species cultures was generally higher than in systems with forced adhesion (Table 3). CNP here increased the growth rate of *C. acnes* and decreased the growth rate of *S. epidermidis* in the system without pre-adhesion. In the second system, *S. epidermidis* growth was accelerated; however, the maximal OD was lower than in the control samples. As the staphylococci were dominative, in systems without forced adhesion, the growth parameters were closer to *S. epidermidis* mono-species cultures, yet slightly higher. In addition, CNP decreased the growth rate of the dual-species community. In systems with forced initial adhesion, in both mono-species cultures, biofilms seemed to be dominative, hence the growth rate was lower than that in the first system. Interestingly, here the hormone had an opposite effect: it stimulated the growth rate of staphylococci and inhibited cutibacteria. Of note, the dual-species communities here grew even faster than the *S. epidermidis* mono-species cultures. This may be a result of the impact of *C. acnes* on the community. In addition, the hormone slightly stimulated the growth of the dual-species community in the same way as for the *S. epidermidis* mono-species cultures. The growth curves are compiled in Figure 7. 

### 3.5. Differential Gene Expression Analysis

Principal component analysis (PCA) and hierarchical clustering were used as methods of quality control with suggested similarity between replicates in RNA-seq (Appendix A). The RNA-seq method showed that studied microorganisms had a great impact on each other. However, the first thing we studied was a gene expression shift in *S. epidermidis* when it was cultivated under anaerobic instead of aerobic conditions (Appendix A). In total, the expression changed in 92 genes, where 83 were upregulated and 9 were downregulated. Interestingly, besides many hypothetical proteins, there were upregulated genes for some membrane transport proteins, alternative electron transport chain proteins such as cytochrome D ubiquinol oxidase subunit II, acetolactate synthase and proteins of drug resistance. Among the downregulated genes, there were genes of 6S rRNA and 16S rRNA, which is logical and potentially due to the generally slower growth of staphylococci under anaerobic conditions. These results seem to be due to a dramatic change in environment and a shift to an anaerobic lifestyle. 

Under anaerobic conditions, the addition of CNP into the medium led to the downregulation of two genes for both hypothetic proteins (55 aa and 39 aa, Table 4). Both proteins are rather similar to DUF3125-containing proteins; both have an unknown function and are not found to be similar to any other proteins. On the other hand, it is unclear why only those proteins were downregulated without any cascade of differential expression reactions. Further studies are needed to explain why no gene was found in *C. acnes* with a changed expression in the presence of CNP, despite the established effect of the hormone on staphylococci. Potentially there was a post-translational effect of CNP.

When mixed, the bacteria changed each other’s gene expression profiles dramatically (Appendix A). In *C. acnes*, the presence of staphylococci in the community led to shifts in the expression of 54 genes, most of which were downregulated. Amongst the downregulated genes were RNA polymerase sigma factor gene, transcription regulators, some stress response proteins and the CAMP factor toxin 1 gene. In addition, four proteins of 50S and 30S ribosome subunits were upregulated (Appendix A). Hence, it could be suggested that, in the presence of staphylococci, *C. acnes* has to shift transcription and translation processes, and *S. epidermidis* also potentially decreases the cytotoxicity of *C. acnes*. In staphylococci, 22 genes were changed in their expression (17 were upregulated and 5 were downregulated). In *S. epidermidis*, the gene for RNA polymerase sigma factor was upregulated, in addition to the lysostaphin resistance protein gene, genes of a number of ABC transporter proteins (including an ATP-binding part of the zinc transporter) and three genes of the CHAP-domain-containing proteins (Appendix A). Therefore, potentially in the presence of *C. acnes*, staphylococci possessed more active transcription and a more resistant cell envelope. Among the downregulated genes were the lipase gene, succinate dehydrogenase gene and mannose-6-phosphate isomerase class I gene. The latter is interesting because the enzyme is zinc-dependent and keeping in mind the upregulation of the zinc ABC transporter, there is potentially competition and a zinc shortage in the community. This could also be proved by the simultaneous upregulation of the zinc ABC transporter in *C. acnes*. Generally, the shifts in gene expression of different ABC transporters in both bacteria in the community suggest competitive interactions, especially for microelements such as zinc. In addition, because of the presence of mostly upregulated genes in staphylococci and mostly downregulated in cutibacteria, this raises the following question: Could we suggest this as additional proof of the dominative role of *S. epidermidis* in the community?

In dual-species biofilms, interactions between microorganisms led to changes in the expression of genes in *C. acnes*, in the presence of CNP, which were not observed in mono-species cutibacteria biofilms (Table 5). In total, five genes were downregulated in cutibacteria inside the community in the presence of CNP. The first was the putative ABC-transporter of TIGR03773. Second, metal permease is close to the zinc uptake transporters (in accordance with the BLAST results, the percentage identity with the zinc ABC transporter of *C. acnes* was 97.38%). Hence, the hypothesis of zinc-oriented interactions provides further evidence. The third gene codes for another membrane protein containing a potentially extracellular M domain. Interestingly, the L28 ribosome assembly protein [35] and the S14 ribosome assembly protein [36] were downregulated in the presence of CNP, while in the control dual-species biofilms, those proteins were upregulated in comparison to mono-species *C. acnes* biofilms.

In staphylococci, in the dual-species community, the genes of two proteins were downregulated in the presence of CNP. The first, called “hypothetic”, was Na-translocating system protein MpsB, which is potentially responsible for cation and dissolved bicarbonate translocation in staphylococci, and potentially in virulence establishment [37]. The second was the gene of NADH-quinone oxidoreductase subunit L—a protein involved in the oxidative phosphorylation chain. Interestingly, the expression of those genes was not affected in the control dual-species community, in comparison with the mono-species *S. epidermidis* biofilms. 

Additionally, to approve or disprove the RNA-seq data, we conducted qPCR for at least five independent repeats for each sample. We decided to also check the results for mono-species biofilms of *S. epidermidis* under anaerobic conditions and for dual-species biofilms. In Table 6, the comparative data of RNA-seq and qPCR are presented for two genes of hypothetical proteins, which were downregulated in mono-species *S. epidermidis* biofilms in the presence of CNP. For the first gene, in only one of the five samples was there downregulation, according to qPCR analysis (Table 6), while in one sample, there was no reaction, and in the remaining three, there was upregulation. Consequently, it is quite difficult to conclude whether the gene was actually downregulated or upregulated. For the second gene, three of the five independent qPCR repeats matched with the RNA-seq data; hence, it seems that this gene, at least, is downregulated more frequently than it is upregulated. Despite this imbroglio, we can establish that both genes differed in their expression, and at least one seems to be downregulated in most cases.

In dual-species biofilms, there were many more differences between qPCR and RNA-seq results (Table 7). In two groups of statistically independent repeats, in the case of *C. acnes*, all five genes were mostly upregulated rather than downregulated, and no case was observed with no reaction. Hence, we suggest that two potential explanations can be proposed: (1) the instability of expression of all those genes led to such data variability; however, all genes are upregulated; (2) the differences occurred due to primer systems specificities in the qPCR and RNA-seq methods. However, in our previous work [27], the same method was conducted, and data for most genes in the RNA-seq method was approved by qPCR. Hence, these genes seem to be unstable in reactions with CNP administration. Nevertheless, all genes were changed in their expression, and we established that those genes were differentially expressed in dual-species communities in the presence of CNP in cutibacteria.

In the case of *S. epidermidis*, qPCR has shown the downregulation of the gene for NADH-quinone oxidoreductase subunit L in five cases out of ten. With the addition of three samples of RNA-seq, we could hypothesize the downregulation of this gene in the presence of CNP. For Na-translocating system protein MpsB, in five samples, qPCR showed triple the downregulation and double the upregulation. Hence, we suggest the downregulation of the gene is approved.

## 4. Discussion

Bacterial communities are very functionally complex structures, even if they consist of only two species. Furthermore, the interactions of species can not only dramatically shift the behavior of neighbors inside the biofilm, but even their susceptibility to different active compounds synthetized by community members or the human organism. In previous works, we found similar phenomena in the action of estradiol on the community of *L. paracasei* and *M. luteus*, where the presence of *M. luteus* in the community led to principal transformation of the inhibitory estradiol effect on lactobacilli to stimulatory [6]. In the community of *S. aureus* and *K. schroeteri*, ANP inhibited the growth of staphylococci in the presence of kytococci, while in mono-species *S. aureus* biofilms, we found no inhibitory effect of ANP in the same concentration.

In the present study, we once again showed that NUPs and CNP in particular affect the communities of staphylococci and cutibacteria. Moreover, as was shown previously, CNP modulates the community in the same way as ANP by giving the advantages to *C. acnes* [17]. In addition, in mono-species *C. acnes* biofilms, there was no effect of CNP on gene expression. Despite this, the tendencies observed allow us to cautiously suggest that CNP may not be a direct metabolic effector in this concentration, in the case of cutibacteria, but it may serve as a molecule that at least modifies some intermediate molecules at the posttranslational level. For instance, it may have some effect on the cell surface properties, as we observed for ANP in our previous work [18]. In the case of *S. epidermidis*, the CNP effect is absolutely clear and well established. In addition, it seems that CNP increased the initial growth of the biofilms and decreased the growth in planktonic cultures; however, it is not clear which exact function(s) may have those hypothetical proteins and which genes were affected. Potentially, those proteins may be a part of an unknown signaling pathway or surface adhesins. If CNP decreases the growth rate of cells, but increases the adhesion ability, this can explain why the planktonic cultures of *S. epidermidis* grow slower, biofilms grow faster, but the final biomass yield after 72 h was lower than that for the control. The adhesin hypothesis may also be strengthened by the fact that the effect of CNP was observed on the hydrophobic surface of the PTFE, and not on the hydrophilic glass. Hence, it is possible that the hydrophobic interactions become more expressed in the presence of the hormone. However, inside the community, another two genes were downregulated in *S. epidermidis*.

A deeper investigation of the mono-species biofilms and the community showed that bacteria have a great impact on each other with regard to gene expression levels. Dozens of genes were upregulated and downregulated in both bacteria, suggesting that each counterpart has a significant effect on the others. Despite *C. acnes* growing slower than staphylococci, the presence of cutibacteria dramatically changes the metabolism of staphylococci. Interestingly, bacteria potentially compete for zinc, and CNP shifted the expression of some zinc-dependent genes in *C. acnes* in a dual-species community. Hence, the mechanism of CNP action may also be connected to zinc metabolism. Of course, we should also remember the obvious but, nevertheless, dramatic shift in the gene expression profile in mono-species *S. epidermidis* biofilms. 

However, qPCR applied to the mono-species *S. epidermidis* biofilms and dual-species biofilms revealed results that were not as easy to interpretate. In mono-species *S. epidermidis* biofilms, there was mostly approval of RNA-seq by qPCR; however, in the community, there was mostly disapproval. The question is therefore: how can we interpret the differences in the results obtained in RNA-seq analysis and qPCR? In the beginning of this study, no analyses of the transcriptome of duals-species communities could be found in the published literature. However, during the last year, several deep and substantial studies were published with two papers dedicated to communities of *Candida albicans* with different bacteria (*Staphylococcus aureus* [38], *Streptococcus mutans* [39]), one manuscript about a fungal community of *C. albicans* and *Cryptococcus neoformans* [40], and one article focused on “coaggregated” *Streptococcus gordonii* and *Fusobacterium nucleatum* in mixed pellets, i.e., not in actual biofilm [39]. Interkingdom communities seem to be easier to analyze; however, in the streptococci-fusobacteria community, there was also a good correlation in results. In all works, authors reach consistency between RNA-seq and qPCR results, and hence, the first thought we had was that there is a mistake in sample preparation or experimental design. Then, several points came to our attention. First, it is important to note which genes were used as a reference. Short et al. [38] and Lu [39] et al. used 16S/18S rRNA genes for *C. albicans*/*S. aureus* [41,42] and *S. gordonii*/*F. nucleatum,* respectively. In case of *C. albicans* [39] and both *C. albicans* and *C. neoformans* [40], actin-encoding act1 gene was used as a reference. Interestingly, Lu and colleagues analyzed only the transcriptome of *C. albicans* and no data for *S. mutans* were presented. Nevertheless, we also used 16S rRNA primers as a reference, so we went the same way as Lu and colleagues. Additionally, we could not make a mistake in all the statistical repeats in 8 to 13 replicates (3–5 independent RNA-seq replicates and 5–10 independent qPCR replicates). Hence, potentially the 16S rRNA gene may not be a proper reference for this biofilm community. According to Sampathkumar and colleagues, sometimes choosing the reference based only on RNA-seq data may be an erroneous approach [43], and additional bioinformatic studies are needed. Additionally, potentially the genes we found are small, have few exons, and are actually expressed lower compared to genes with consistent expression measurements between the two methods, according to another study [44]. Such “non-concordant” genes, as it was described by prof. Coenye in his recent discussion article [45], represent a smaller part of the genome, and may show reproducible non-consistent results between two methods. Additionally, in the case of such genes, the time-fold change is frequently lower than 2 in modulo [45]. 

Such a mismatch could be explained by the fact that most genes studied were “non-concordant”, as was described by Prof. Coenye in his recent discussion article [45]. However, all of “our” genes had a time-fold change of ˃2 in RNA-seq experiments, and mostly 1 < x < 2 in qPCR experiments. It is possible that the primer system may not have been appropriate; however, not all the genes were fully mismatched in qPCR in comparison with RNA-seq, while the same 16S rRNA reference genes were used in all the repeats. In addition, in our previous work conducted in the same way with *M. luteus*, this problem was less acute, at least for 7 of the 11 genes with differential expression revealed via RNA-seq [27]. In addition, in mono-species biofilms of *S. epidermidis* in the present study, qPCR mostly approved the RNA-seq data. Thus, a potential explanation may lay in the general complexity of interactions between species in the community and potential “non-concordance” of the genes with differential expression. Nevertheless, we can be sure that (i) we found the genes that change their expression in the presence of CNP in both bacteria, and (ii) we established the effect of CNP on this community as regulatory, and it seems that the key target for CNP is *S. epidermidis*. Lastly, (iii) is based on the following facts: (a) the staphylococcal biofilms are sensitive to the hormone with statistically significant differences when the bacterium is cultivated in mono-species cultures and (b) staphylococci are a dominative part of the community due to their more intensive and fast growth and competitive advantage over *C. acnes*. The full mechanism of CNP action might be solved in future research, which should include (i) proteomics investigation; (ii) a full description of all hypothetical proteins in *S. epidermidis* including which genes were differentially expressed in the presence of the hormone, and their functions; (iii) if possible and appropriate, making the specific mutant (genetically engineered) strains with alterations in genes with CNP-affected differential expression. The same is applicable to *C. acnes*. 

As a result, we determined the effect of CNP in a physiological blood plasma concentration on *S. epidermidis* mono-species biofilms, and on the community of *S. epidermidis* and *C. acnes*. Despite the fact that the effect of the hormone is little manifested externally, its presence causes significant changes inside the cells, especially in cutibacteria. This allows us to assert the universality of the phenomenon of NUP regulatory effects on human (including skin) commensal microbiota and gives us a wide perspective for future research.

## 5. Conclusions

Considering the results obtained in the present study together with previous works, the role of human natriuretic peptides as microbial community regulators seems to be established. Although in monospecies biofilms *C. acnes* seems to be indifferent to CNP, the presence of *S. epidermidis* radically shifts the behavior of cutibacteria, revealing the target genes in cutibacteria sensitive to CNP. This shows the complexity of microbial interactions even in the simplest dual-species communities. Additionally, the proposed competition for zinc (based on the differentially expressed genes) allows to suggest the importance of this microelement to both bacteria and its involvement in the CNP regulation of biofilms. Despite the partial mismatch between RNAseq and qPCR results, the genes sensitive to the presence of CNP were reliably established, and the further investigation of CNP-mediated regulation in both bacteria will shed the light on this aspect of microbial endocrinology, including its medical and industrial applications.

## Figures and Tables

**Figure 1 biology-12-00436-f001:**
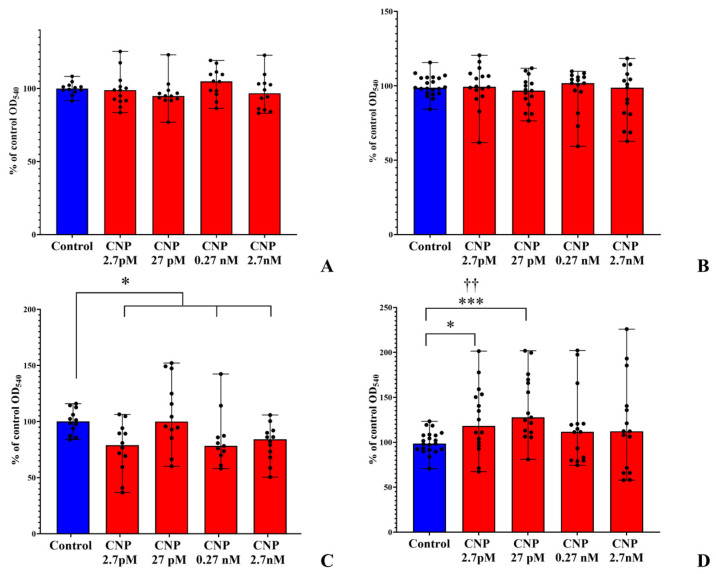
Effects of different CNP concentrations on planktonic cultures and biofilms of *S. epidermidis* on PTFE cubes. (**A**), 24 h planktonic cultures; (**B**), 72 h planktonic cultures; (**C**), 24 h biofilms; (**D**), 72 h biofilms. Symbols: * represents *p* < 0.05; *** represents *p* < 0.001; †† represents *q* < 0.005; absence of daggers represents *q* ˃ 0.05.

**Figure 2 biology-12-00436-f002:**
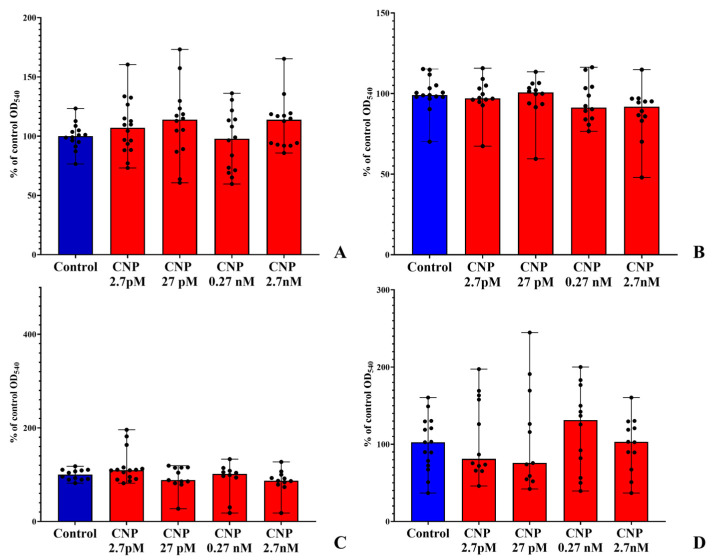
Effects of different CNP concentrations on planktonic cultures and biofilms of *C. acnes* on PTFE cubes. (**A**), 24 h planktonic cultures; (**B**), 72 h planktonic cultures; (**C**), 24 h biofilms; (**D**), 72 h biofilms.

**Figure 3 biology-12-00436-f003:**
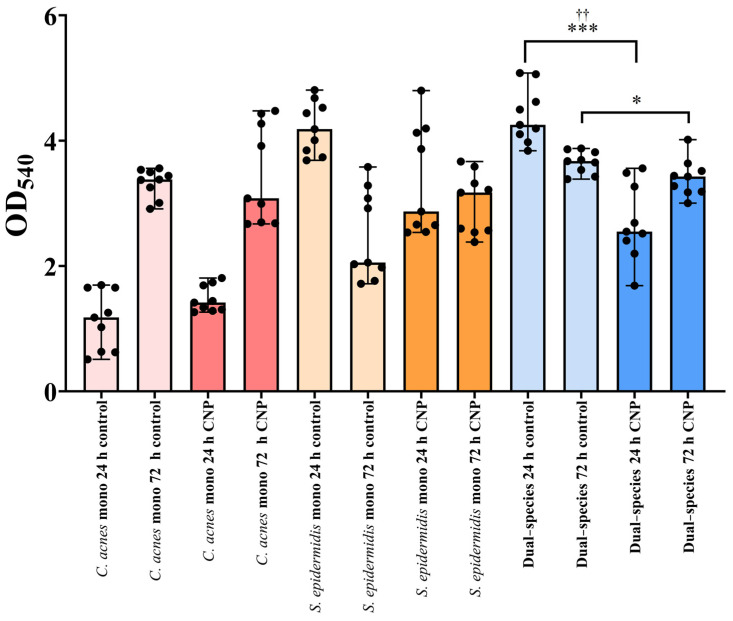
MTT staining of mono-species and dual-species biofilms of *C. acnes* and *S. epidermidis* on GMFFs. Symbols: * represents *p* < 0.05; *** represents *p* < 0.001; †† represents *q* < 0.005; absence of daggers represents *q* ˃ 0.05.

**Figure 4 biology-12-00436-f004:**
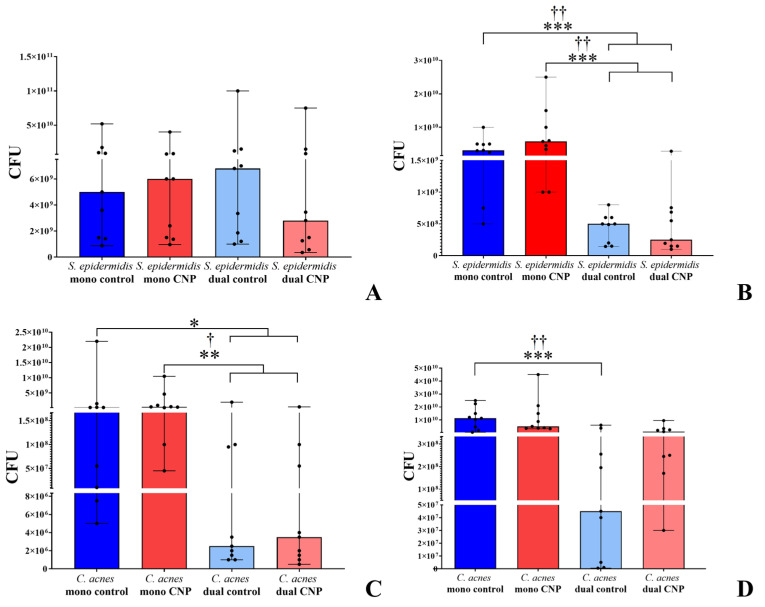
CFU counts in mono-species and dual-species biofilms of *C. acnes* and *S. epidermidis*. (**A**), CFU counts of *S. epidermidis* after 24 h of incubation; (**B**), CFU counts of *S. epidermidis* after 72 h; (**C**), CFU counts of *C. acnes* after 24 h of incubation; (**D**), CFU counts of *C. acnes* after 72 h of incubation; * represents *p* < 0.05; ** represents *p* < 0.01; *** represents *p* < 0.001. †† represents *p* < 0.005; † represents *q* < 0.05; absence of daggers represents *q* ˃ 0.05.

**Figure 5 biology-12-00436-f005:**
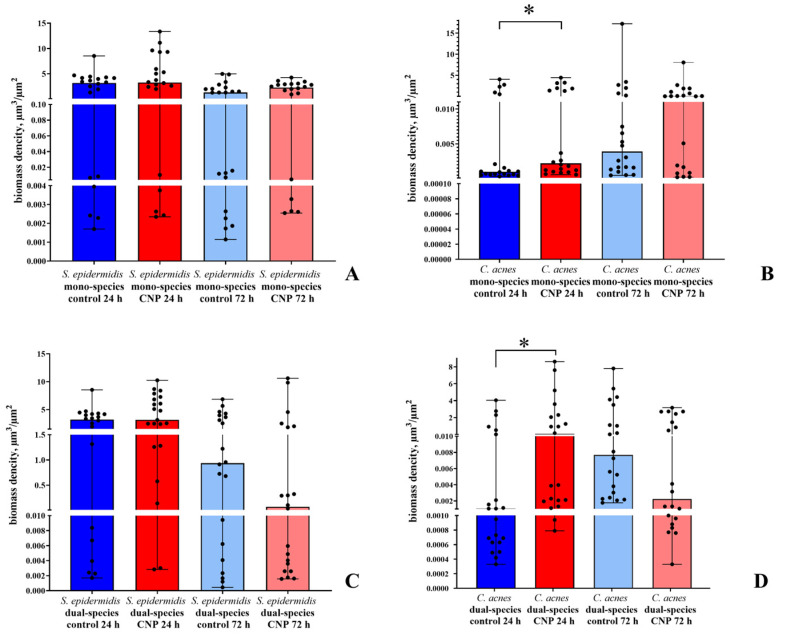
The analysis of CLSM images of mono-species and dual-species biofilms. (**A**), mono-species biofilms of *S. epidermidis*; (**B**), mono-species biofilms of *C. acnes*; (**C**), *S. epidermidis* proportion in dual-species biofilms; (**D**), *C. acnes* proportion in dual-species biofilms; * represents *p* < 0.05.

**Figure 6 biology-12-00436-f006:**
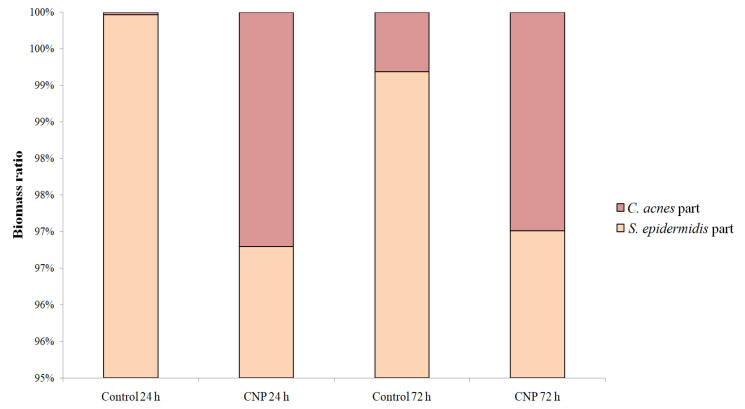
The ratio of *S. epidermidis* and *C. acnes* biomass in dual-species communities, analyzed using CLSM.

**Figure 7 biology-12-00436-f007:**
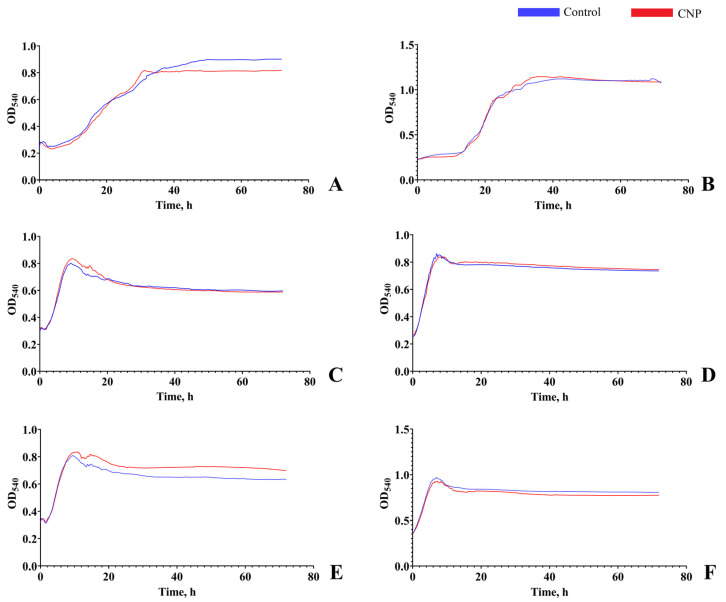
The growth curves for mono-species and dual-species cultures of *S. epidermidis* and *C. acnes*. (**A**,**C**,**E**), in the system with forced initial adhesion; (**B**,**D**,**F**), in the system without forced initial adhesion. (**A**,**B**), mono-species *C. acnes*; (**C**,**D**), mono-species *S. epidermidis*; (**E**,**F**), dual-species biofilms.

**Table 1 biology-12-00436-t001:** Programs and parameters for read processing.

The Program	Version	Parameters Applied in the Study ^1^
Trimmomatic	0.39	ILLUMINACLIP: [adapters file path]:2:30:10:1: TRUE TRAILING:3 SLIDINGWINDOW:4:15 AVGQUAL:20 MINLEN:30
BWA	0.7.17	
Salmon	1.3.0	--libType A --gcBias --numGibbsSamples 20
sva	3.38	
DESeq2	1.22.2	

^1^ Those parameters simultaneously (1) differ from defaults and (2) have a significant impact on the program application results.

**Table 2 biology-12-00436-t002:** PCR mix composition.

№	Reagent	V, µL
1	dNTP, 2.5 мM	2.5
2	10× PCR buffer B	2.5
3	MgCl_2_, 25 мM	2.5
4	Primer mix, 10 pM/µL	2
5	SynTaq DNA polymerase, 5 U/µL	0.5
6	dd H_2_O	15
7	DNA matrix	0.2

**Table 3 biology-12-00436-t003:** Kinetic parameters of mono-species and dual-species cultures in different model systems.

		Without Pre-Adhesion			With Pre-Adhesion		
		Growth Rate, h^−1^	Generation Time, h	Maximal OD_540_	Growth Rate, h^−1^	Generation Time, h	Maximal OD_540_
*S. epidermidis* mono-species	Control	0.25	2.8	0.86	0.17	3.94	0.8
	CNP	0.22	3.19	0.84	0.2	3.51	0.84
*C. acnes* mono-species	Control	0.11	6.58	1.12	0.1	7.44	0.9
	CNP	0.13	5.42	1.14	0.09	7.74	0.82
Dual-species	Control	0.18	3.85	0.97	0.18	3.75	0.82
	CNP	0.16	4.23	0.93	0.21	3.32	0.84

**Table 4 biology-12-00436-t004:** Differential expression of *S. epidermidis* biofilms under anaerobic conditions in the presence of CNP, revealed via RNA-seq.

Locus Tag	Protein Description	Log2 (Expression Level Ratio ^1^)	Standard Error of log2 (Expression Ratio)	*p*-Value	*q*-Value
B6C95_03235	Hypothetical protein	−4.02808	0.600497	4.59 × 10^−7^	0.000553
B6C95_04760	Hypothetical protein	−7.89188	1.204172	1.04 × 10^−8^	2.52 × 10^5^

^1^ The expression ratio was shown as a ratio of CNP-treated samples to control samples, here and in the following text.

**Table 5 biology-12-00436-t005:** Differential expression in dual-species biofilms in the presence of CNP, revealed via RNA-seq.

Locus Tag	Protein Description	Log2 (Expression Level Ratio)	Standard Error of log2 (Expression LEVEL Ratio)	*p*-Value	*q*-Value
*C. acnes*					
HMPREF9571_ RS01535	TIGR03773 family transporter- associated surface protein	−4.54224	0.606570082	5.22661 × 10^−9^	1.3051 × 10^−5^
HMPREF9571_ RS02495	Metal ABC transporter permease	−3.45326	0.607693267	5.41364 × 10^−5^	0.02703573
HMPREF9571_ RS06980	Choice-of-anchor M domain- containing protein	−3.83189	0.543910001	1.92397 × 10^−7^	0.00018143
HMPREF9571_ RS11165	50S ribosomal protein L28	−10.5324	1.839060929	2.17973 × 10^−7^	0.00018143
HMPREF9571_ RS11175	30S ribosomal protein S14	−4.65725	0.720441212	3.84643 × 10^−7^	0.00024011
*S. epidermidis*					
B6C95_06200	Hypothetical protein (Na-translocating system protein MpsB)	−5.14976	0.986638	2.6 × 10^−5^	0.031301
B6C95_06205	NADH-quinone oxidoreductase subunit L	−4.89638	0.922791	2.42 × 10^−5^	0.031301

**Table 6 biology-12-00436-t006:** Comparison of RNA-seq and qPCR results in *S. epidermidis* mono-species biofilms under anaerobic conditions.

RNA-Seq			qPCR				
Locus Tag	Protein Description	Log2					
B6C95_03235	Hypothetical protein	4.02808	no reaction	12.1	1.8	−1.3	1.3
B6C95_04760	Hypothetical protein	7.89188	−1.4	5	1.2	−3.3	−1.7

**Table 7 biology-12-00436-t007:** Comparison of RNA-seq and qPCR results in dual-species biofilms under anaerobic conditions.

RNA-Seq			qPCR									
Locus Tag	Protein Description	Log2 (Expression Level Ratio)	May 2022				September 2022					
			1	2	3	4	1	2	3	4	5	6
*C. acnes*												
HMPREF9571_ RS01535	TIGR03773 family transporter-associated surface protein	−4.54224	3	2.3	−1.5	1.6	2.9	3.3	−2.9	1.6	−8.3	2.3
HMPREF9571_ RS02495	Metal ABC transporter permease	−3.45326	5.4	1.8	−1.3	1.7	26.4	4.2	−3	1.5	−33	1.9
HMPREF9571_ RS06980	Choice-of-anchor M domain-containing protein	−3.83189	4.9	1.4	−1.5	2	16.7	3.8	−1.8	2	−20	3.5
HMPREF9571_ RS11165	50S ribosomal protein L28	−10.5324					2.7	3.3	−2.3	1.9	−12.5	1.6
HMPREF9571_ RS11175	30S ribosomal protein S14	−4.65725	3.9	1.6	−1.5	1.3	7.8	2.9	−2.5	no reaction	−8.3	1.9
*S. epidermidis*												
B6C95_06200	Hypothetical protein (Na-translocating system protein MpsB)	−5.14976						12.1	2.6	−25	−1.6	−100
B6C95_06205	NADH-quinone oxidoreductase subunit L	−4.89638	−5.5	−5.5	2.9	1.5	2.4	no reaction	−1.8	1.9	−3.6	−25

## Data Availability

All reads produced for this study were uploaded to the NCBI Sequence Read Archive (SRA) and are available under the project PRJNA899449.

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
