# Peer review of "C-Type Natriuretic Peptide Acts as a Microorganism-Activated Regulator of the Skin Commensals Staphylococcus epidermidis and Cutibacterium acnes in Dual-Species Biofilms"

_biology, 2023, doi:10.3390/biology12030436_

Round 1
Reviewer 1 Report
The manuscript is scientifically interesting and well written. Phenomenon of NUP regulatory effects on human commensal microbiota is an important issue in the long term.
Documentation of biofilm inhibition is proposed to be additionally supported with photos after the use of microscopic techniques, e.g. scanning electron microscope or using photos from other microscopic techniques (see the publication: Suppression of Staphylococcus Aaureus biofilm formation andvirulence by a benzimidazole derivative, UM-C162, Scientific Reports, 2018). Microscopic image analysis is an important part of microbiological documentation. Selected samples with the greatest inhibitory effect can be documented.
Author Response
Dear Reviewer,
Thank you for your remarks. Concerning the microscopy images, we prefer to use a confocal microscopy approach to study biofilms. Thus, all the 3D images were presented at Supplementary materials, figure 1. Therefore, all the samples with the most remarkable stimulation or inhibition effects were presented as you have proposed. It is also important to keep in mind that for instance EM is not an “universal” approach for biofilm studies due to the method specificities. And sometimes there is a risk of artifacts.
Hence, the use of EM is the potential way to perform the more focused studies in future, potentially to find the special cell forms (i.e. persisters etc.) if there are some premises. In this study the were not an aim to search of such a cell forms in biofilms. The aim was just to find out any effect of CNP and find some preliminar potential ways of its action.
With best regards,
Andrei
Reviewer 2 Report
The manuscript by Maria et al describes the regulation of Staphylococcus epidermidis and Cutibacterium acnes in biofilms. A lot of studies have been done to compile the manuscript. I suggest the following additions/correction before the publication of the manuscript.
1. There are several typos and mistakes in the manuscript. I have highlighted a few of them in the attached file.
2. Authors should cite some of the literature reported to manage staphylococci infections such as
https://www.sciencedirect.com/science/article/abs/pii/S0223523421002518
https://doi.org/10.1016/j.ejmech.2021.113402
3. Standard deviation in some of the graphs is extremely high. Authors should give some explanation.

Author Response
Dear Reviewer,
Thank you for your remarks. We will answer them point by point.
- We addressed the marked places in the text and corrected the typos.
- We added the text in the Introduction section (see the lines 49-55). We addressed the mentioned article and also some other references.
- We used the plot “Median with range” in the GraphPad, and all the values were plotted on the graphs. Hence, in this regime, there were no standard deviation, because error bars were plotted from the minimal value to the maximal value in each bar. We decided to use this way of depicting of the results because we (i) used mostly non-parametric Mann-Whithney criterion, so it is more suitable here. And (ii) this is more demonstrative method to depict all the values as they are in experiments.
We add all the necessary parts in the text, so please find the new variant of the manuscript.
With best regards,
Andrei